# The Future for the Children of Tomorrow: Avoiding Salt in the First 1000 Days

**DOI:** 10.3390/children11010098

**Published:** 2024-01-14

**Authors:** Giorgia Mazzuca, Silvia Artusa, Angelo Pietrobelli, Giuseppe Di Cara, Giorgio Piacentini, Luca Pecoraro

**Affiliations:** 1Pediatric Clinic, Department of Surgical Sciences, Dentistry, Pediatrics and Gynecology, University of Verona, 37126 Verona, Italy; giorgia.mazzuca@studenti.univr.it (G.M.);; 2Department of Pediatrics, University of Perugia, 06129 Perugia, Italy

**Keywords:** salt intake, hypertension, obesity, salt sensitivity, salt taste, children, first 1000 days of life

## Abstract

It is widely known that optimal nutrition in the first 1000 days of life positively impacts the child’s development throughout adulthood. In this setting, salt should not be added to complementary feeding. In developed countries, salt intake is generally higher than recommended for children. Excessive salt intake is the major determinant of hypertension and is associated with several cardiovascular outcomes. Therefore, pediatricians have a key role in raising awareness among parents to avoid salt consumption in the first 1000 days of life to ensure better health for their children. Starting from a review of the literature published in PubMed/MedLine regarding the short- and long-term consequences of salt consumption during the first 1000 days of life, our comprehensive review aims to analyze the beneficial effects of avoiding salt at such a vulnerable stage of life as the first 1000 days. Obesity, hypertension, increased salt sensitivity, high sweet drink consumption, increased mortality, and morbidity persisting in adult age represent the principal consequences of a higher salt intake during the first 1000 days of life.

## 1. Introduction

### 1.1. Why Are the First 1000 Days of Life Important?

The period encompassing the moment of conception to the age of two years represents the initial 1000 days of life, and it is critical for lifelong health and well-being and represents a crucial time window for several preventive interventions [1,2]. The most relevant period for neurological development, characterized by rapid changes, occurs during the first 1000 days. Fundamental primary brain structures and processes form during that period (i.e., sensory system, hippocampus, myelination, and monoamine neurotransmitter systems) [3]. Indeed, these days are considered the most significant period for brain and body growth, for acquiring language and sensory pathways for hearing and vision, and for developing cognitively higher functions. It represents a time of huge neurodevelopment opportunities and great vulnerability during which children are more susceptible to environmental stimulations [2,3,4].

### 1.2. Salt Consumption: Neither Depletion nor Excess

Both a chronic depletion and an excess of salt intake could be dangerous. A deficiency of sodium, especially in newborns, is responsible for growth retardation and alteration in neurodevelopment. Furthermore, iodine deficiency emerges among the potential effects of reducing sodium intake in the diet. Iodinated salt is critical for the synthesis of thyroid hormones, and its inadequate consumption represents the most common cause of preventable impairments in mental health worldwide [5]. Several factors, particularly nutrition, may contribute to developing long-term salt-related consequences later in life, such as obesity, cardiovascular, endocrine, and metabolic diseases [6]. Hence, proper complementary foods are important for nutritional and developmental needs [1]. Complementary feeding is fundamental for ensuring an adequate intake of nutrients and avoiding excess intake of calories, unhealthy fats, sugars, and, especially, salt [4].

High sodium is typically found in common table salt, meat, milk, shellfish, bread, crackers, and snacks. Also, many food condiments, such as fish sauces and soy, are rich in sodium [7]. However, although most food has low natural iodine content, food from the sea, especially seaweeds, includes varying amounts of iodine. In addition, the primary sources of iodine are also found in other nutrients, such as milk products (derived from cow, camel, and goat), iodine-fortified salt, and iodine-rich water [8].

Despite this, previous studies on the Italian pediatric population have shown a high prevalence of excessive salt consumption and inadequate iodine intake [9].

### 1.3. Recommendations for Salt Intake during Childhood

The European Society for Paediatric Gastroenterology, Hepatology, and Nutrition (ESPGHAN) and the World Health Organization (WHO) established the recommended salt intake during childhood for infants 0–6 months old. Recommended salt consumption is based on sodium levels in breast milk. For older infants between 6 and 24 months, adequate salt intake is determined by sodium content in breast milk and complementary food [10], avoiding adding salt during meal preparation [11]. Finally, for children aged 24 months to 15 years, the WHO recommends adjusting the maximum sodium levels of intake (2 g/day) according to their energy requirements relative to those of adults [12,13,14].

### 1.4. Main Consequences of High Salt Consumption

The key concern regarding salt consumption during the first 1000 days is preventing and mitigating adverse outcomes, particularly hypertension and obesity in short- and long-term life [15]. The main blood pressure (BP) determinants are total peripheral vascular resistance and cardiac output. The mechanisms implicated in regulating BP by acting on these two factors could be divided into short-term (modulation of the autonomous nervous system activity) and long-term mechanisms (the actions of endocrine factors such as the renin–angiotensin–aldosterone system and vasopressin) [5]. Moreover, previous evidence has explained the pathophysiological mechanism underlying the link between increasing salt consumption and the associated risk of developing obesity: high salt consumption can cause increased sugar-sweetened drink consumption by stimulating thirsty sensations, and obesity is strongly related to the large intake of foods rich in salt and calories [16]. However, it has also been hypothesized that high sodium intake may be a risk factor for developing obesity, regardless of energy intake [16]. Finally, a reduction in salt consumption determines a reduction in sweet food consumption, with positive effects on blood pressure and body weight [17]. Pediatricians could know the salt content in the main average foods used in the first 1000 days (Table 1) and play a pivotal role in ensuring proper nutrition throughout a child’s life from prenatal care, including maternal nutrition during pregnancy, to postnatal care, encompassing growth monitoring and education for children and parents [15]. Our comprehensive review aimed to analyze the beneficial effects of avoiding salt intake during the first 1000 days of a child’s life, carefully considering the possible detrimental repercussions that excessive salt consumption might have on developing lifelong disorders.

## 2. Physiology of Sodium Chloride

The extracellular fluid is composed of sodium (Na^+^), the main cation, and chloride (Cl^−^), the main anion. Sodium chloride is commonly known as salt, which is composed of 40% sodium and 60% chloride, and it exerts several important functions [5,7]: it controls the osmolarity of the extracellular fluid; regulates body water and fluids balance; controls the acid–base balance and excitation of nerve and muscle cells; and releases digestive secretions and absorbs several nutrients (i.e., water, glucose, amino acids) [5].

## 3. Salt Intake: Benefits and Consequences

### 3.1. Benefits

#### 3.1.1. The Role of Sodium in Newborns and Infants’ Growth

As explained above, sodium is fundamental for fluid balance [19]. A positive sodium balance is necessary for newborns and infants’ development of the skeleton and growth stimulation [19]. Experimental studies demonstrated that chronic sodium depletion, especially in preterm and low-birth-weight infants, in whom tubular immaturity determine sodium loss, results in negative salt balances, hyponatremia, and growth retardation [19]. Hence, sodium supplementation is important in that population during the first 1000 days of life. Supplementation of 4–5 mmol/kg of sodium daily should be used in preterm infants to lose less body weight postnatally and regain birth weight more rapidly. On the contrary, 1 mmol/kg daily of sodium is sufficient for adequate growth in full-term infants fed through breastfeeding [19].

#### 3.1.2. The Importance of Iodized Salt

Another condition that should be avoided, particularly in children, is the inadequate intake of iodinated salt [9]. Iodine is a critical mineral nutrient and a component of thyroid hormones, contributing to their synthesis. Thyroid hormones (fT4, fT3) play a key role in proper central nervous system function, neurocognitive and brain development, and healthy growth, regulating overall metabolism [12,13]. Iodine deficiency is the world’s most common cause of preventable mental disability [20,21]. The iodine status of pregnant and reproductive-age women is recognized as a crucial risk factor [20,21]. In many parts of the world, the amount of iodine intake must be increased to ensure physiological needs. Thus, the World Health Organization, to minimize that condition, proposed the utilization of iodine-enriched salt, with an iodine concentration between 20 and 40 mcg/g, even while maintaining a sodium intake of less than 5 g/day or below [9]. Several measures have been instituted to encourage the consumption of iodine-rich salt throughout the entire country (i.e., modifying dietary patterns and food production techniques, avoiding veganism and some forms of vegetarianism, using iodized salt, changing industry and agricultural practices, and increasing the intake of dairy products) [20,21]. However, many specific populations and areas still require iodine supplementation. Further investigation is necessary to determine whether modified or novel normative measures should be considered to raise iodine intake [20,21].

### 3.2. Short-Term Consequences

#### 3.2.1. The Development of Flavor and Taste

Considerable attention is given to transitioning from an exclusive milk diet to the diet consumed in early childhood, especially in the first 1000 days of life [11]. Newborns naturally prefer salty and sweet tastes and have an innate aversion to bitter tastes typical of fruit and vegetables [4]. The kinds of food that children initially either prefer or reject could be advantageous in a setting where energy and mineral-dense foods are scarce. However, it could be harmful in the context of increasing obesity [22]. The WHO has no recommendations regarding food tastes or acceptance, encouraging different food experiences to transition to family foods properly [4]. The preference for salt may also be attributable to postnatal events [23,24,25,26]. Flavor and taste preferences determined at the beginning of life continue through childhood and adolescence, underlining how fundamental early nutrition is to establish priorities for healthy food and beverages throughout life [4,17].

#### 3.2.2. How to Regulate Predisposition to Salty Taste

By acting on earlier experiences, it is possible to modify those predispositions and, thus, minimize health problems during adult life [11,22]. Moreover, salt preference appears to be more variable than the preference for sweet food. Exposure to flavors (salt, sweet, and bitter) in amniotic liquid, breast milk, formula, and solid foods could be helpful for infants to develop healthy food preferences [22]. Despite the exposure during pregnancy or breastfeeding, repeated exposure to different foods is important to ensure their acceptance [4]. Salt taste identification seems to develop later, probably at 2–6 months, not in the neonatal stage [18,19]. Infants prefer significantly greater salt concentrations than adults, and adding salt to food may promote their higher intake later in life. The range of food flavors is passed from the maternal diet to amniotic liquid and breast milk to promote the mothers’ healthy diet during pregnancy and breastfeeding [22]. Repeated exposure and diet variety help both breastfed and formula-fed weaned infants easily accept nutrients. These foods should be part of the family diet so that once the preference is developed, the child will continue to be exposed to the food to retain the flavor [22]. Exploring whether an optimal time in which experience promotes better food appreciation would be an important step in encouraging healthy nutrition and, in turn, facing the numerous diseases associated with poor dietary and food decisions [22].

### 3.3. Long-Term Consequences

To date, it is recognized that such a sensitive period of life as the first 1000 days can strongly influence the future health of the child/adolescent. The high consumption of salt intake can have several negative consequences later in life, mainly including hypertension and obesity [2,11].

#### 3.3.1. Hypertension

Throughout the last century, salt has been the focus of scientific research concerning its impact on blood pressure and the cardiovascular system, representing a leading cause of mortality worldwide [20,21]. It is well-known that elevated salt intake is one of the principal risk factors for developing hypertension in adults [27]. Less evidence is available among children and adolescents, and the results are controversial [27]. Excessive salt intake can lead to the same complications as in adults: volume expansion and arterial hypertension [19].

##### Pathophysiology of Hypertension

Hypertension appears to result from the interaction between genetic, environmental, behavioral, and epigenetic factors [17]. The mechanism determining the gradual rise in blood pressure and arterial alterations, both structural and functional, represents a long and complex process that seems to mature from conception to adulthood [23]. The “fetal programming”, via both direct and indirect effects of salt intake during pregnancy, could be considered the ensemble of epigenetic mechanisms through which arterial hypertension develops. This prenatal condition seems to determine a higher oxidative stress, endothelial dysfunction, and a reduction of capillary density, promoting the development of hypertension [17]. According to Suckling et al., consumption of high-sodium-rich food could determine a transitory rise in plasma sodium level, with adverse effects on the vascular system, which are mediated by reactive oxygen species. It determines an increase in arterial stiffness, hypertrophy of vascular smooth cells, sympathetic nervous system, and renin–angiotensin–aldosterone system; a reduction in nitric oxide production and the renal dopaminergic system; and increased BP [28,29].

##### Hypertension and Nutritional Intervention

A blood pressure ≥ 95th percentile for age, sex, and height is defined as hypertension in the pediatric population [10]. A meta-analysis and a systematic review of observational and pilot studies by Leyvraz et al. highlighted the association between salt intake and BP. Nutritional interventions, characterized by an average duration of 16 weeks and an average decrease in salt intake of 1.2 g, showed a reduction in systolic pressure of 0.6 mmHg and a reduction in diastolic pressure of 1.2 mmHg [10]. Moreover, Leyvraz et al. demonstrated that adding 1 g of sodium daily increases systolic pressure by 0.8 mmHg and diastolic pressure by 0.7 mmHg [27]. Other systematic reviews have shown no significant differences in systolic or diastolic BP in children aged 1 to 18 who underwent a short-term sodium reduction intervention [30].

##### Sodium Sensitivity

Furthermore, the BP response to sodium in the diet is generally supposed to differ greatly among people, and “sodium sensitive” is described as a large BP response to sodium intake changes, categorizing individuals into salt-sensitive and insensitive groups [5,10]. Children born small for gestational age (SGA) and those born with low birth weight (<2.5 kg defined according to WHO) have an increased risk of developing high BP after elevated sodium intake, with 37% and 47% increase in salt sensitivity, respectively [10]. Limiting salt intake from the first 1000 days of life aims to prevent the development of hypertension in adulthood, a major risk factor for cardiovascular disorders [31]. Tempestive recognition and lifestyle modification management are crucial to minimize adverse outcomes [31].

#### 3.3.2. Obesity

##### Epidemiology

Body mass index (BMI) > 2 SDS above the median WHO guidelines is considered obesity; BMI > 1 SDS and ≤2 SDS of the median weight-for-height is considered overweight [2]. Obesity and overweight in children and adolescents are a major public health challenge. It has been dramatically increased during the past few years, even among children. [15]. This condition is associated with metabolic complications [32]. The World Health Organization estimates that 5.6% of children and adolescents globally were obese or overweight in 2016, accounting for about 340 million of these individuals between the ages of 5 and 19 [15,33].

##### The Relationship between Obesity and Salt Intake

An imbalance in energy intake and expenditure is the primary cause of obesity in children and adolescents [2]. Recent studies have focused on the association between salt intake and pediatric obesity, examining the underlying pathophysiological mechanisms [10]. Previous studies have hypothesized that a high salt intake could determine an increasing consumption of sugar-sweetened drinks by stimulating thirsty sensations. Moreover, poor-quality diets, such as processed food commonly consumed by children and adolescents, are characterized by high salt, calories, sugar, and fat. The coexistence of all these conditions can lead to the development of obesity. However, in their observational study, Yuan et al. analyzed how a high salt content could contribute to obesity independently of the abovementioned mechanisms [16]. High salt consumption is associated with an augmented extracellular water volume, resulting in a small increase in body weight < 1 kg and a direct increase in fat mass [16]. Furthermore, a daily increase of 1 g of salt was associated with a 28% rise in the incidence of overweight and obesity in children and a rise of 0.73 kg in body fat and 0.44 kg in lean mass [16].

##### Obesity Prevention Strategies

Given the great influence that obesity has on children and adolescents’ overall health, it is crucial to prevent it. In this setting, the first 1000 days of life represent a unique time for obesity prevention by supporting a healthy and balanced diet for both parents at the time of conception period and pregnancy and an accurate monitoring of the baby’s growth by encouraging breastfeeding until six months of age and by introducing complementary food between four and six months of age, avoiding the addition of sugar, sugar fluids, and, especially, salt to the diet [2]. Moreover, fruit and vegetable consumption should be introduced early. Animal protein should be limited to reduce the risk of adiposity rebound, and essential fatty acid intake should be promoted in the diet [2].

## 4. Effects of Low-Sodium Diet

Several dietary components contain sodium, and cultural settings and habits represent the major contribution of the population’s sodium intake [7]. Common table salt, processed food (i.e., bread, crackers, and snacks), and natural food (i.e., milk, meat, and shellfish) generally contain high sodium. Thus, a diet rich in processed food and low in fresh fruit and vegetable consumption increases the risk of developing hypertension and other diseases [7]. The association between salt consumption and blood pressure, blood lipids, catecholamine levels, and any adverse consequences in children was examined by a meta-analysis. Reduced salt consumption, specifically, results in a significant decrease in resting diastolic blood pressure of 0.87 mm Hg (0.14 to 1.60 mm Hg) and resting systolic blood pressure of 0.84 mm Hg (0.25 to 1.43 mm Hg) [7]. The effects of reduced sodium intake on blood lipids, catecholamine levels, or negative consequences in children were not discussed in any research [7]. A useful strategy to limit the consumption of high-sodium-rich food is to incorporate low-sodium salt substitutes into the children’s diet, with products with less sodium than regular salt replaced with potassium or other minerals [34]. Strategies involving potassium-enriched salt alternatives as the primary option may differ per country based on dietary preferences and sodium sources in the population’s diet [35]. Potassium is a necessary mineral and the primary cation in the intracellular fluid. The cellular interaction between potassium and sodium is fundamental for fluid equilibrium. Using potassium-enriched, reduced-sodium salt as a substitute for regular table salt (sodium chloride) represents an important approach for reducing dietary sodium and increasing dietary potassium to promote health [36]. Consuming food high in potassium, at recommended levels, is still a key component of a healthy diet, and it is associated with improvements in the quality of the overall diet. Moreover, potassium-enriched nutrients seem to determine a reduction of BP and the risk of cardiovascular disorders [36]. Recent trials have demonstrated that the systolic and diastolic BP reduction is most likely not due only to less salt consumption but also to an increase in potassium intake [37].

Uncertainty has been raised concerning the likelihood that potassium-enriched salt replacements increase potassium blood levels in vulnerable people (i.e., those with renal failure) [38]. Although the risk associated with hyperkaliemia might not be worth the benefits to blood pressure and the cardiovascular system, some researchers have proposed warning labels for these products [39].

## 5. Salty and Sweet: Two Health Enemies

In the past, salt and sugar, elements of important nutritional value, were difficult to obtain, generating a marked, innate preference for salty and sweet tastes over bitter flavors and flavors characteristic of most vegetable tastes [4]. In addition, they promote food conservation, developing numerous recipes and preparations involving their use [17]. Despite the natural propensity for salt and sweet tastes, lifestyle modifications and eating habits have determined a higher use of these nutrients. Children’s consumption of salt and sugar is commonly considered to be through separated and different foods [17]. However, several studies demonstrated a strong relationship between salt, fluids, and consumption of sweet drinks [40]. Daily salt consumption has been shown to have a significant positive relationship with BMI, which resulted in considerably higher in hypertensive children than in normotensive ones [32,41]. Kos et al. study corroborates other studies’ results, demonstrating that hypertensive children are most commonly obese or overweight, whereas normotensive ones have normal BMI [32]. Furthermore, hypertensive children, compared to normotensive peers, were characterized by a higher percentage of fat mass and by a significant correlation between fat mass and systolic and mean arterial BP [32,42,43]. A decrease in salt consumption may lead to a decrease in sweet drink consumption, resulting in positive effects on BP and body weight [17].

For this reason, it is essential to establish a preventive program for families and industries aimed at reducing salt and sugar consumption [17,44]. In the pediatric age, the most emerging phenomenon that requires intervention is overweight and obesity. All intervention projects intended to improve children’s lifestyles by promoting proper nutrition, with an indication to increase the intake of vegetables and fruit and to reduce the addition of salt and the consumption of sweet drinks [17]. These goals make close collaboration between pediatricians, family members, schools, and, on a more far-reaching level, food industries necessary [17].

## 6. Discussion

To date, great emphasis has been given to the proper use of salt intake, especially during the first 1000 days of life, a time of great importance for cognitive and physical development, during which the rate of brain growth is the greatest of all the stages of life. Particular nutrients can have positive and negative consequences depending on the duration and dose of exposure [4]. That process is complicated and involves different biological, cellular, and structural changes in the brain through a specific succession [3]. Timing is critical.

### 6.1. Factors Influencing Neurobehavioral Development

An optimal and healthy neurobehavioral development requires that all essential factors be present at their biologically defined moments [3]. Several conditions influence this process. Maternal nutrition during the prenatal period and infant feeding in the first 1000 days of life emerge among them. Children and adult health risk factors (i.e., hypertension and obesity) are influenced by the proper consumption of micro- and macro-nutrients during that period. They could be programmed through nutritional management [3]. The consequences of undernutrition include considerable mortality and morbidity, retardation in motor development, and increased risk of noncommunicable diseases in later life [4]. Specifically, iodine deficiency during pregnancy and early childhood can compromise neurodevelopment and growth, increase infant mortality, and raise the incidence of thyroid disorders [45]. Despite its positive and beneficial effects, salt intake can also have negative consequences. Consumption of salt is considered a well-established risk factor for obesity and hypertension, regardless of body weight, gender, and age [16]. According to recent research, a low-salt diet can help prevent hypertension, particularly in overweight children, premature and SGA newborns, and those with hypertension or non-preventable cardiovascular risk factors (such as family history of hypertension) [16]. Consumption of salt intake early in life is associated with an increased risk of hypertension during childhood and elevated BP in adulthood. Moreover, it determines a higher consumption of sweet-rich foods and influences taste development. Therefore, it is crucial to raise awareness among parents about this issue [17]. The underlying mechanisms begin very early in life despite cardiovascular disorders occurring mostly in adulthood. Hypertension and obesity are the most important long-term consequences of inadequate salt consumption. These long-term illnesses could have both health and socioeconomic implications for society. Reducing salt intake benefits children and adults [46].

### 6.2. The Role of a Low-Salt Diet for Public and Individual Health

Even though a positive sodium balance is essential for brain growth and neurocognitive development during the early years of life, a low-salt diet in older children has the same cardiovascular-protective effects as in adults. Given the harmful effects of hypertension and obesity, it is critical to strengthen public health and individual initiatives to raise health risk awareness and offer education on healthier eating attitudes and lifestyles to prevent these emerging disorders from a very early age [32]. It would be necessary to implement efforts to minimize salt consumption, given the significance of early prevention at the individual and public health levels [3]. Pediatricians and other health care providers are responsible for educating and informing parents about the importance of proper nutrition for children. This awareness program should start as soon as possible, from the prenatal period, focusing on maternal, fetal, and neonatal nutrition to ensure normal neurodevelopment and to avoid the negative consequences of poor and inadequate diet, especially during the first 1000 days of life and with a particular concern about salt intake [3]. Several studies have highlighted that the percentage of sodium intake derived from natural foods is low (5–10%). In contrast, about 70–80% of sodium intake occurs through processed foods (e.g., bread) and 10–15% by adding salt during meal preparation [47]. Strategies to promote the reduction of salt intake are different. It is fundamental to ensure a structured nutrition education program aimed at improving environmental conditions, making healthier choices easier and more accessible, making nutrition labeling mandatory, and acting on legislative changes in the salt distribution system. Furthermore, it is necessary to provide appropriate health claims and work on legislative taxation changes. Combined with greater family education and awareness, these intervention strategies could be successful [5].

### 6.3. Strengths and Limitations

Inherent limitations, such as author subjectivity bias and lack of a rigorous and systematic literature search, flaw the narrative design of this review. This might inadvertently result in incomplete coverage of the literature and limited generalizability of the search outcomes.

## 7. Conclusions

The first 1000 days of life are extremely delicate, especially regarding proper brain development and healthy growth. What influences these aspects most is nutrition from the prenatal stage during pregnancy. In this context, adequate intake of macro- and micronutrients plays a fundamental role. More attention has been given to the use of salt in the first 1000 days of life. Salt plays a key role in infants (especially low-birth-weight and premature infants) for skeletal development and smooth growth. Therefore, its chronic deficiency could have detrimental effects. Also relevant is iodized salt deficiency, the primary risk factor for thyroid and neurological issues in young people. There is compelling evidence linking high blood pressure to excess salt intake. Hypertension is the most effective risk factor for developing cerebrovascular, cardiovascular, and renal disorders. In addition, improper salt intake may also be associated with the development of obesity and overweight in children and adolescents. In conclusion, given the health implications that excess salt intake can cause, pediatricians or any children’s health providers should make families aware of this issue, educating parents and caregivers about the strategies to reduce their dietary sodium intake and to contribute to minimizing the health and social effects of this condition. Further studies are required to reinforce the narrative review with a systematic approach to strengthen the base and draw more solid conclusions.

## Figures and Tables

**Table 1 children-11-00098-t001:** Salt content in the main average foods used in the first 1000 days.

Food Product Category	Sodium Density (mg/1000 Kcal)—Mean	Sodium Concentration (mg/100 g)—Mean
Vegetables	501 (302–700)	20 (12–28)
Fruit	77 (59–96)	5 (4–5)
Cereals	31 (12–51)	12 (5–20)
Mixed grains and fruit	103 (61–146)	9 (5–13)
Savory snacks	1382 (1114–1649)	486 (367–604)
Juices/drinks	184 (150–218)	9 (7–10)
Dairy-based desserts	421 (357–484)	42 (31–53)
Dry grain-based dessert	399 (301–496)	169 (125–214)
Dry fruit-based snacks	383 (302–463)	138 (108–167)
Cereal bars and breakfast pastries	744 (558–930)	248 (199–298)
Dinners, soups, and vegetables	503 (428–578)	28 (25–31)

Modified from [18].

## Data Availability

Not applicable.

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
