# Peer review of "The Future for the Children of Tomorrow: Avoiding Salt in the First 1000 Days"

_children, 2024, doi:10.3390/children11010098_

Round 1

Reviewer 1 Report

Comments and Suggestions for Authors

Comments on the Quality of English Language

Please look at my pdf document.

Author Response

Thank you very much for providing me the opportunity to review this manuscript. While reading the manuscript, I could make the following comment from my side:

Summary: The critical importance of optimal nutrition during the first 1000 days of life for a child's lifelong development is well-established. This period significantly influences adult outcomes. Specifically, the addition of salt to complementary feeding is discouraged due to its potential negative impact. Despite this guidance, developed countries commonly exceed recommended salt intake for children, leading to increased risks of hypertension and various cardiovascular issues. Pediatricians play a crucial role in educating parents about the detrimental effects of excessive salt consumption during the initial 1000 days of life. This comprehensive narrative review aims to assess the positive outcomes associated with avoiding salt during this vulnerable developmental stage, emphasizing the need for heightened awareness to ensure better long-term health for children.

We thank the reviewer for comprehending the significance of our narrative review.

Issues:

Writing a narrative review on the critical importance of optimal nutrition during the first 1000 days of life offers several benefits, such as- synthesis of existing knowledge on the impact of nutrition during the early developmental stages. It enables a contextual understanding of the challenges related to salt intake in developed countries, emphasizing the real-world implications of nutritional guidelines. It also allows for a holistic examination of the role of pediatricians in educating parents, offering insights into the practical aspects of healthcare delivery and public awareness. By exploring the literature, it can identify gaps in current knowledge and areas where further research is needed, guiding future scientific inquiries. It helps in effectively communicating the critical importance of avoiding excessive salt consumption during the initial 1000 days of life to ensure better long-term health for children. The review contributes to understanding the public health implications of salt intake, emphasizing the need for heightened awareness and potential interventions to address this health concern. It may inform policy discussions by presenting a consolidated view of the evidence, potentially influencing guidelines and recommendations related to child nutrition. It can also serve as an educational tool for healthcare professionals, parents, and policymakers, fostering a broader understanding of the relationship between nutrition, salt intake, and long-term health outcomes. However, while writing a narrative review on the importance of optimal nutrition during the first1000 days of life is valuable, there are potential demerits associated with this approach: ï‚· Bias and Subjectivity: Narrative reviews can be susceptible to the biases and subjectivity of the author. Without a systematic and transparent methodology, there is a risk of presenting information in a way that aligns with the author's preconceived notions or beliefs. ï‚· Limited Rigor: Unlike systematic reviews or meta-analyses, narrative reviews may lack the rigorous methodology required to minimize bias and ensure a comprehensive evaluation of existing literature. This can impact the reliability and validity of the conclusions drawn. ï‚· Incomplete Coverage: A narrative review may not cover all relevant studies systematically, leading to the omission of critical information or a skewed representation of the existing literature. This could result in an incomplete understanding of the subject matter. ï‚· Difficulty in Replicability: The lack of a standardized methodology makes it challenging for others to replicate the review process. This can hinder the verification of the findings and limits the overall scientific rigor of the study. ï‚· Limited Generalizability: It may focus on specific studies or populations, limiting the generalizability of its findings. This can be a drawback when attempting to apply the conclusions to broader contexts or diverse populations. ï‚· Risk of Cherry-Picking Evidence: Authors might unintentionally or intentionally select studies that support a particular viewpoint, neglecting contradictory evidence. This can lead to a biased representation of the literature and an incomplete understanding of the topic. To mitigate these demerits, it is advisable to complement narrative reviews with systematic approaches, ensuring transparency in the review process, and acknowledging potential biases. Additionally, authors should emphasize the need for further research to strengthen the evidence base and draw more robust conclusions.

Thank you very much for your appreciation of our manuscript and the insightful comments. Although a systematic review would have certainly been characterized by a more solid and rigorous methodology, the narrative design of our article was agreed upon with the editor for this Special Issue. While having present the inherent limitations of such a study design, we have tried to conduct a search as comprehensive as possible and to summarize the available evidence using a wide breadth approach. According to your helpful comments, we have mentioned the limitations highlighted here in the Discussion section.

Reviewer 2 Report

Comments and Suggestions for Authors

This is an interesting review article with adequate quality. Some points should be addressed.

- The abstract should contain a bit more information about the methods used and the main results and conclusions of this review.

- The Introduction setion should be split into at least 3 paragraphs.

- The authors should report (maybe in the introduction section that what foods contain sodium and what foods contain iodine and whether their levels are adequate in a typical diet in both infants and adults.

- Again, section "3.1 Benefits" should be split into 2 or 3 more small paragraphs.

-  Again, section "3.2 Short-term consequences" should be split into 2 or 3 more small paragraphs.

-Again, sections "3.2.1 Hypertension" and 3.2.2 "Obesity" should be into 2-3 smaller paragraphs in order to be more easily understood for the readers.

- The interesting statement in lines 228-230 could further be analyzed by adding more information about it.

- The Discussion section should be split into 3-4 smaller paragraphs.

- The main strengths and the main limitations of the currently available clinical studies about the impacts of sodium intake should be reported and discussed at the "Discussion section".

- In the last few years, there are certain subsitutes of classical salt in the market which contain lower levels of sodium but higher levels of potassium. This issue should be reported by the authors and should by scrutinized by them. Is it more suitable for the infants? Are there any studies for such substitutes either in infancy or in adolescence and adulthood. 

Author Response

This is an interesting review article with adequate quality. Some points should be addressed.

- The abstract should contain a bit more information about the methods used and the main results and conclusions of this review.

We thank the reviewer, we have implemented the abstract according to your suggestions.

- The Introduction setion should be split into at least 3 paragraphs.

Thank you very much for your comment. We have divided the introduction into four sections.

- The authors should report (maybe in the introduction section that what foods contain sodium and what foods contain iodine and whether their levels are adequate in a typical diet in both infants and adults.

We thank the reviewer for the suggestion. We have reported the additions requested in the introduction section.

- Again, section "3.1 Benefits" should be split into 2 or 3 more small paragraphs.

We kindly thank the reviewer. We have divided “3.1 Benefits” into two sections.

-  Again, section "3.2 Short-term consequences" should be split into 2 or 3 more small paragraphs.

Thank you very much for your comment. We have divided the “3.2 Short-term consequences” into two sections.

-Again, sections "3.2.1 Hypertension" and 3.2.2 "Obesity" should be into 2-3 smaller paragraphs in order to be more easily understood for the readers.

We thank the reviewer. We have divided the 3.2.1 Hypertension" and 3.2.2 "Obesity" into three sections respectively.

- The interesting statement in lines 228-230 could further be analyzed by adding more information about it.

We thank the reviewer. We have updated the lines 228-230 with more informations.

- The Discussion section should be split into 3-4 smaller paragraphs.

Thank you very much for your comment. We have divided the Discussion into three sections.

- The main strengths and the main limitations of the currently available clinical studies about the impacts of sodium intake should be reported and discussed at the "Discussion section".

Thank you very much for your comment. We have added the Discussion with strengths and limitations of the study.

- In the last few years, there are certain subsitutes of classical salt in the market which contain lower levels of sodium but higher levels of potassium. This issue should be reported by the authors and should by scrutinized by them. Is it more suitable for the infants? Are there any studies for such substitutes either in infancy or in adolescence and adulthood.

We thank you the reviewer for the suggestion. We have explored this topic analyzing the most recent studies.

Round 2

Reviewer 2 Report

Comments and Suggestions for Authors

The authors have significantly improved their manuscript. From my point of view, this is a state-of-the art critical review article of high scientific quality.

Author Response

I thank the reviewer for his/her comment.